# Identification of Non-Traumatic Bone Marrow Oedema: The Pearls and Pitfalls of Dual-Energy CT (DECT)

**Giovanni Foti** [1,*] , **Gerardo Serra** [2] , **Venanzio Iacono** [3] , **Stefania Marocco** [4] , **Giulia Bertoli** [4] , **Stefania Gori** [5] **and Claudio Zorzi** [3]

1 Department of Radiology, IRCCS Sacro Cuore Don Calabria Hospital, 37042 Negrar, Italy
2 Department of Anesthesia and Analgesic Therapy, IRCCS Sacro Cuore Don Calabria Hospital, 37042 Negrar, Italy; Gerardo.Serra@sacrocuore.it
3 Department of Orthopaedic Surgery, IRCCS Sacro Cuore Don Calabria Hospital, 37042 Negrar, Italy; venanzio.iacono@Sacrocuore.it (V.I.); claudio.zorzi@sacrocuore.it (C.Z.)
4 Centre for Tropical Diseases, IRCCS Sacro Cuore Don Calabria Hospital, 37042 Negrar, Italy; stefania.marocco@sacrocuore.it (S.M.); giulia.bertoli@sacrocuore.it (G.B.)
5 Department of Oncology, IRCCS Sacro Cuore Don Calabria Hospital, 37042 Negrar, Italy; stefania.gori@sacrocuore.it
\* Correspondence: giovanni.foti@sacrocuore.it; Tel.: +39-0456013874

**Abstract:** Dual-energy computed tomography (DECT) is an imaging technique widely used in traumatic settings to diagnose bone marrow oedema (BME). This paper describes the role of DECT in diagnosing BME in non-traumatic settings by evaluating its reliability in analyzing some of the most common painful syndromes. In particular, with an illustrative approach, the paper describes the possible use of DECT for the evaluation of osteochondral lesions of the knee and of the ankle, avascular necrosis of the hip, non-traumatic stress fractures, and other inflammatory and infectious disorders of the bones.

**Keywords:** computed tomography; bone marrow; avascular necrosis; inflammation; infection

## 1. Introduction

Bone marrow oedema (BME), arising from bone marrow capillary wall damage, causes an increase in intraosseous pressure and irritation of the sensory nerves [1]. Upon imaging, BME is typically diagnosed by an increased water content that can be detected by fat-suppressed magnetic resonance imaging (MRI) pulse sequences [1]. BME is the hallmark of several non-traumatic painful diseases, including vitamin D deficiency, osteoporosis, microtrauma, chronic liver failure, diabetes mellitus, peripheral arterial disease, and bone marrow oedema syndrome [2,3]. In addition, BME may manifest in several other non-traumatic pathological conditions including osteochondral lesions (OCL), avascular necrosis (AVN), subchondral fractures, and stress fractures [4–6].

MRI is the most frequently used imaging tool in the diagnosis of BME [1]. However, due to contraindications and increased costs, it is not always suitable or available. DECT can suppress normal bone by using a virtual non-calcium (VNCa) algorithm, thus allowing the identification of BME in traumatic and non-traumatic settings [6–17]. Specifically, DECT can be used to diagnose such non-traumatic pathological conditions as gout. In particular, DECT is very useful in atypical presentations, which pose a diagnostic challenge and distinguish it from other arthropathies and masses, such as septic arthritis, rheumatoid arthritis, osteoarthritis, pseudogout, or tumor, especially in sites that are not directly accessible for fluid aspiration [17].

DECT have been used for the evaluation of lymphoproliferative disorders, and chronic inflammatory disorders. It can also be used to evaluate ligaments and menisci [14–17]. Furthermore, DECT applications can reduce metal-induced artifacts and/or increase intrinsic

tissue contrast. This paper describes the role of DECT imaging in diagnosing non-traumatic BME and associated imaging findings.

## 2. Imaging Protocol

A dual-source CT scanner was used (Somatom Definition Flash or Force, Siemens Healthcare, Forchheim, Germany) with tube voltages set at 80/140 kV for tube A and 80/150 kV for tube B, using a tin filter. For imaging of the appendicular skeleton, scanned regions were located at the center of the DE FOV, and the predefined tube current-time product was set at a ratio of 1.6:1 (tube A, 220 mAs; tube B, 138 quality reference mAs) with a detector collimation of $32 \times 0.6$ mm, and pitch of 0.6. An increase of tube-current time-product values up to 300 and 228 mAs was applied, as required, for axial skeleton scans, in cases of increased anatomical thickness, so as to reduce photon starvation artifacts.

## 3. Imaging Interpretation

DECT colour-coded images were reconstructed from soft-tissue kernel (Qr32) images to provide a better signal-to-noise ratio of bone window images [3]. DECT 3D and 2D images were evaluated with standard CT images, using bone and soft tissue windows, on a dedicated offline workstation, to allow adequate windowing and post-processing. DECT 3D images, coding BME in shades of green and normal bone in blue, are generally more sensitive in depicting BME than 2D images are. Further to this, 3D images give an overview of the whole anatomical area. DE-specific information was superimposed onto conventional gray-scale morphological images (thickness, 1 mm; increment, 1 mm) for 2D images. Regardless of the color lookup table adopted, BME is usually coded using colors corresponding to densities between $-30$ HU and 30 HU. However, when assessing non-traumatic BME here, DECT images have been visualized with superimposed color-coded maps only when density values are above a $-50$ HU cut-off. This approach may prove helpful in distinguishing between mild and severe BME. Dedicated windowing can be carried out by increasing or decreasing the level of superimposition of color-coded images. By increasing the level of superimposition, it is possible to detect areas of less obvious BME, whereas by decreasing the visualization of a superimposed BME map, it is possible to evaluate fine anatomical details of underlying bones.

In cases of bone sclerosis, which is relatively common amongst elderly individuals suffering from degenerative osteoarthritis, spared bone is typically coded in violet and BME in shades of green (Figure 1). As bone sclerosis can yield false positive findings [3,9], it is important to be aware of typical bone sclerosis sites so as to reduce the risk of misdiagnosis. It is also useful to compare symptomatic and asymptomatic sides, and to look for any asymmetry. Moreover, it is also recommended to progressively increase the threshold for BME and/or the level of superimposition of BME of bone window images, in order to obtain a clearer depiction. In selected cases, the quantitative assessment of DECT numbers, by using a ROI for questionable areas, can help to achieve a differential diagnosis. However, thresholds may vary widely depending on the anatomical area, age of the patient, and qualitative assessment of DECT images [3].

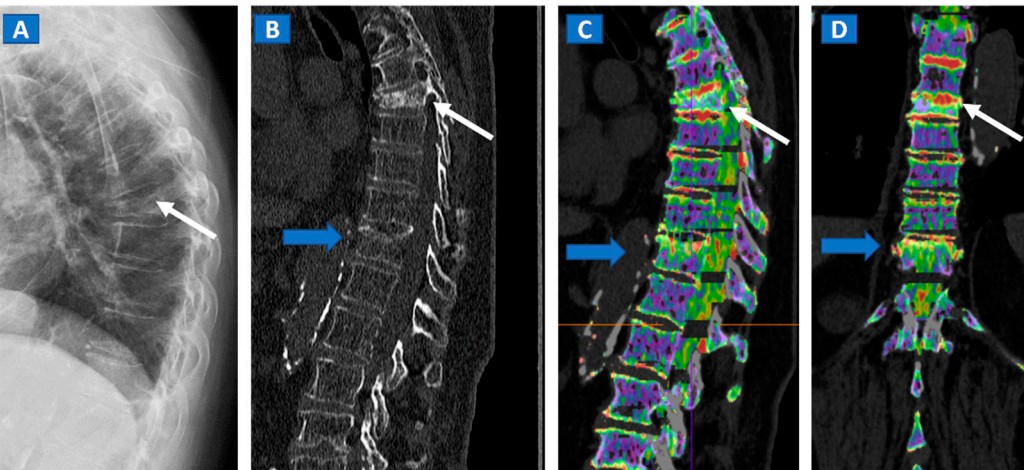

**Figure 1.** An 82-year-old female, unable to undergo MRI, with non-traumatic dorsal vertebral compression fractures. On the sagittal standard radiograph (**A**) a mid-dorsal spine fracture can be seen (white arrow). On the high resolution sagittal 1 mm reconstructed CT image (**B**), an additional fracture of the D9 vertebral body is suspected (blue arrow). On the corresponding sagittal (**C**) and coronal DECT superimposed map (**D**), D5 and D9 fresh vertebral fractures are confirmed by the presence of BME, coded in shades of green (arrows), while normal bone is coded in violet.

## 4. Radiation Burden and Target Diseases

By using tube current modulation software (CARE dose 4D; Siemens Healthcare), the radiation burden is negligible for the appendicular skeleton [2,6,7] and acceptable for the axial skeleton [8,9]. It is important to note that BME incidence and prevalence tend to increase with age due to an increased likelihood of osteoarthritis and subsequent cartilage loss, and because of osteopenia and reduced bone mineralization. Moreover, complex painful syndromes and chronic disorders are more frequently diagnosed in elderly patients, who are often unsuitable candidates for MRI. Importantly, BME can be better identified amongst this population due to progressive decreases in bone density as a result of the relative increase in yellow bone marrow, as compared to red.

The presence and distribution of BME, together with the anatomical changes depicted by standard high-resolution CT images, are fundamental in ascertaining a correct diagnosis and choosing the most appropriate therapeutic approach [2]. Furthermore, concerns regarding radiation exposure are arguably less pressing for elderly patients.

## 5. Vertebral Compression Fractures

DECT can distinguish fresh from old vertebral compression fractures by demonstrating the presence of BME [11]. In a recent metanalysis evaluating 13 studies using MRI as reference for diagnosis, including 515 patients and 926 acute fractures, DECT overall sensitivity was 86.2% with a specificity of 91.2% and accuracy of 89.3% [18]. In addition, DECT can accurately determine the position of hardware inserted during spinal surgery, by reducing metallic artifacts. The high intrinsic contrast of DECT images also assists in determining a differential diagnosis, such as in cases of disc herniation (Figure 2).

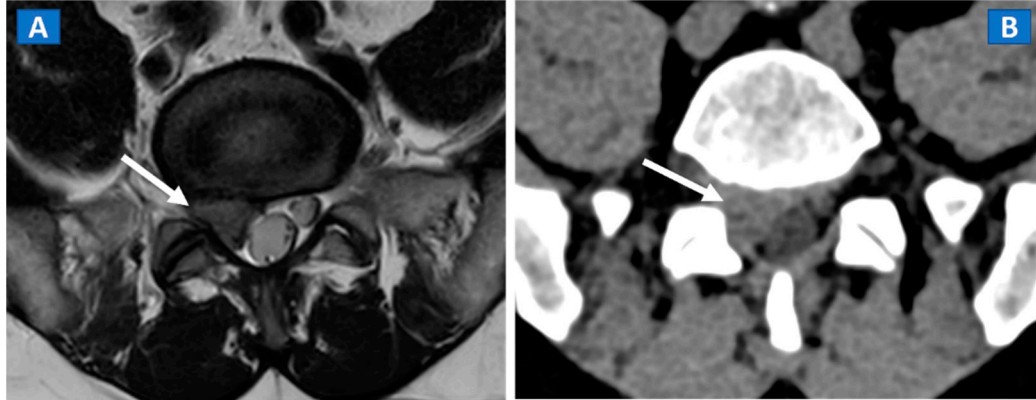

**Figure 2.** A 46-year-old male with non-traumatic, acute right sciatica pain. A large, right postero-lateral disk herniation (arrow) can be recognized at L5-S1 level both on the axial T2-weighted MR image (**A**) and on the corresponding axial 1 mm reconstructed CT image with a soft tissue window (**B**).

## 6. Non-Traumatic Osteochondral Lesions

DECT has been successfully used to diagnose OCL of the knee and ankle both in traumatic and non-traumatic settings [3,7]. Painful OCL are typically associated with BME (Figure 3), often appearing as markedly edematous subchondral areas in DECT images, surrounded by milder and peripherally degrading BME [3,7]. However, some OCL can be missed by DECT, especially in the presence of subchondral sclerosis [8]. The analysis of additional associated findings, such as articular space narrowing that is typical of patients with osteoarthritis or meniscal or ligamentous injuries, may help in the identification of smaller, less obvious BME foci (Figure 4). In such cases, color-coding spared bone in violet and BME in shades of green, and correctly adjusting the level of superimposed BME information are key to an accurate diagnosis.

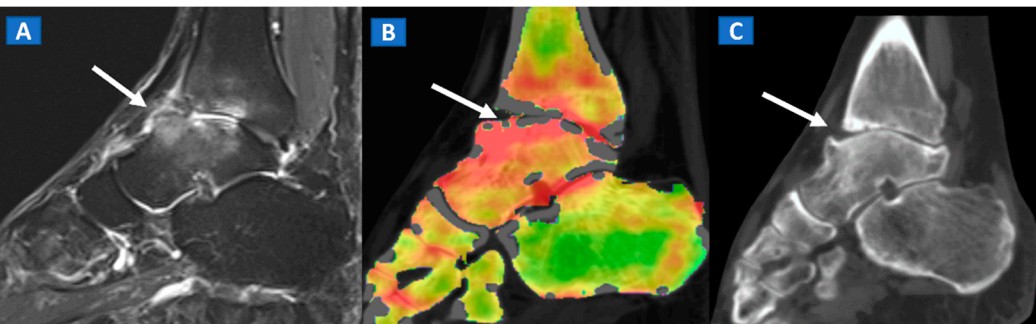

**Figure 3.** A 48-year-old male with OCL of the talar dome and distal tibia. On the sagittal STIR MRI image (**A**), subchondral marrow lesions, associated with thinning articular cartilage, are depicted on the talar dome and distal tibia (arrow). On the corresponding sagittal 1 mm DECT map (**B**), BME is depicted in red (arrow), whereas normal bone is shown in shades of green-yellow. On the sagittal 1 mm reconstructed CT image with a soft tissue window (**C**), it is possible to identify an astragalic beak, subchondral sclerosis and articular space narrowing (arrow) as being the probable causes of non-traumatic BME.

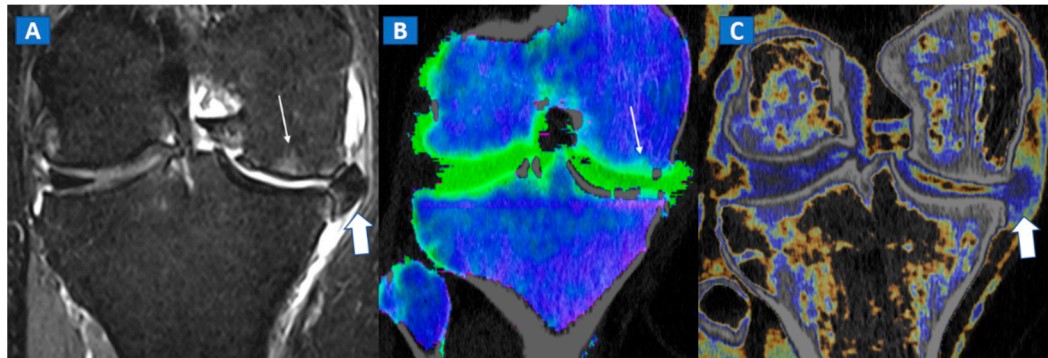

**Figure 4.** A 65-year-old female with non-traumatic medial-sided knee pain. On the coronal STIR MRI image (**A**) the body of the medial meniscus is ruptured and medially dislocated (thick arrow). Tiny, subtle BME areas can be seen on the femoral condyle (thin arrow) due to advanced chondropathy. Tiny areas of BME (thin arrow) are also depicted on the corresponding coronal 1 mm reconstructed DECT image (**B**). By using alternative color-coding (**C**), DECT allows the depiction of the ruptured meniscus (thick arrow), providing a good correlation with MRI findings.

## 7. Avascular Necrosis

The diagnosis of osteonecrosis, defined as the ischemic death of bone, is based on the presence of a double line sign, BME, and subchondral fractures at MRI. With standard high-resolution CT images available (Figure 5), DECT can been used to identify BME and associated imaging findings, as well as to reliably diagnose AVN of the femoral head [9]. Importantly, DECT images can reliably identify subchondral fractures and cortical bone involvement, which are key in the prognosis and management of AVN of the femoral head (Figure 5). Thanks to high resolution images, DECT could be used for staging AVN, depicting for example cystic and sclerotic radiographic changes in early phases, or BME around sub-chondral collapsed areas. CT is also a highly reliable imaging tool in evaluating articular narrowing and calcific loose bodies.

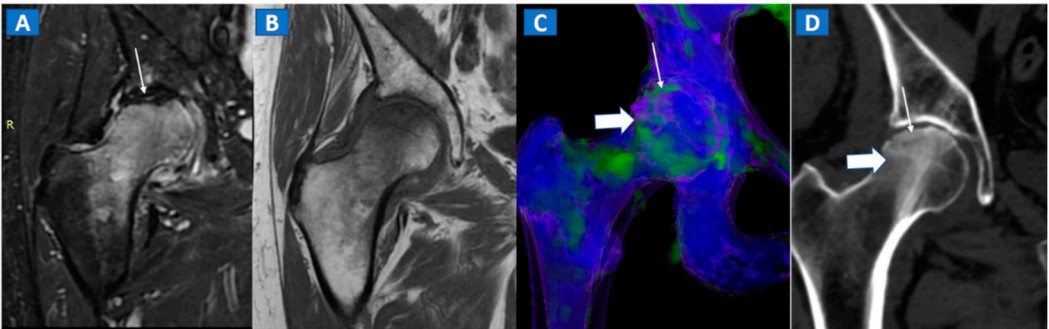

**Figure 5.** A 58-year-old female with AVN of the right femoral head. On the coronal STIR MRI image (**A**), a hypointense serpiginous subchondral line, evident as a double line sign (arrow), is associated with diffuse BME involving the femoral neck and trochanteric region. On the corresponding T1-weighted image (**B**), hypointense BME seems to spare the medial aspect of the femoral head. On the DECT coronal 3D image (**C**), a flattened subchondral edematous area is depicted, with sclerosis-related artifacts hindering clear imaging of any BME of the head (thick arrow), although BME is apparent on the femoral neck. On the corresponding 1 mm coronal CT image with a soft-tissue window (**D**), it is possible to depict an increased density corresponding to the BME distribution (thick arrow), and a subtle subchondral hypodense line, which would be consistent with the diagnosis of a subchondral fracture (thin arrow). R = right.

## 8. Transient Bone Marrow Oedema Syndrome

Transient bone marrow oedema syndrome is a self-limiting disease characterized by pain and localized BME, which often affects lower limb joints. Besides confirming the

tendency of BME to spare subchondral areas (Figure 6), DECT imaging can also rule out any associated findings to narrow the differential diagnosis. DECT is indeed capable to diagnose stress fractures, early AVN stages, and inflammatory arthropathies.

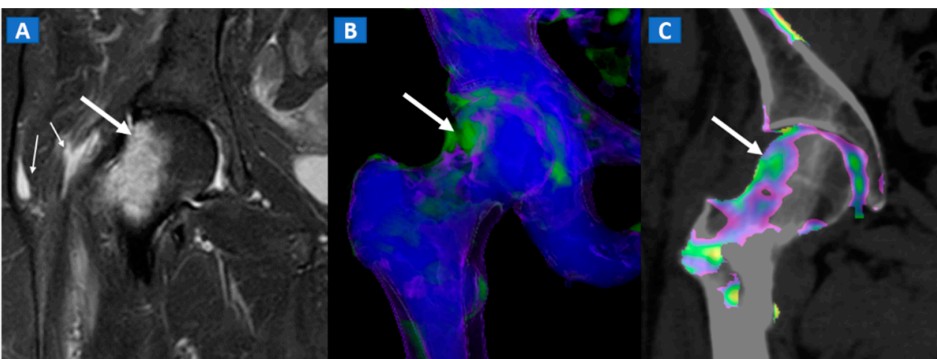

**Figure 6.** A 46-year-old female with non-traumatic right hip pain. On the coronal STIR MRI image (**A**), hyperintense BME of the femoral neck is depicted (thick arrow), sparing the subchondral femoral head area. Note the associated trochanteric bursitis (thin arrows). The pattern of distribution of BME (arrow) is confirmed on the corresponding DECT coronal 3D image (**B**), and on the 1 mm coronal DECT image (**C**) (arrow).

## 9. Stress or Insufficiency Fractures

Stress fractures are caused by repetitive micro-traumatic injuries over time. Insufficiency fractures are a type of stress fracture commonly seen in osteoporotic women, resulting from normal stresses on abnormal bone. As for traumatic non-displaced fractures, DECT can identify stress fractures by depicting BME around the fracture line. High resolution morphological CT images can also be used to finely depict the course of a fracture line, and to distinguish fresh from old fractures depending on the presence of a reactive osteoblastic response (Figure 7).

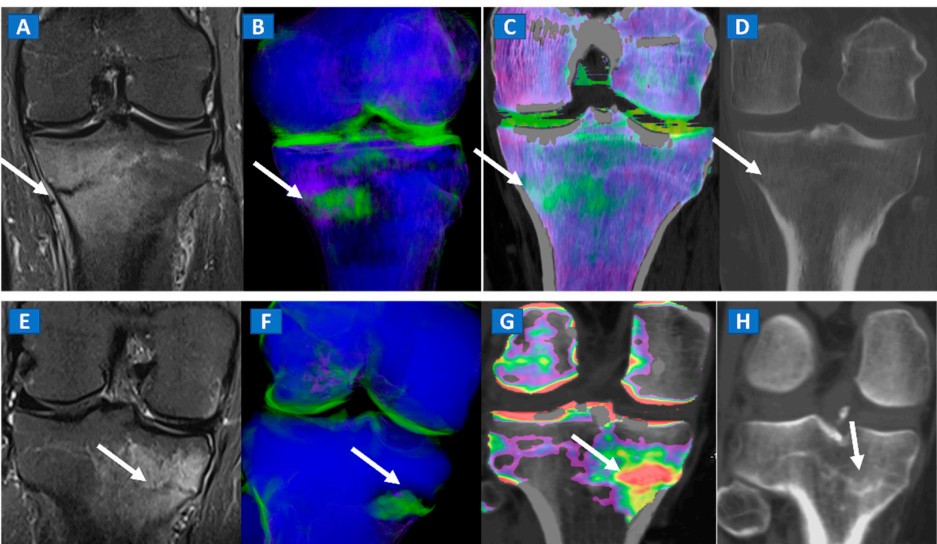

**Figure 7.** A 42-year-old male long-distance runner with a bilateral tibial stress fracture. On the coronal STIR MRI images (**A**,**E**), a hypointense serpiginous subchondral line (arrows), surrounded by subtle BME involving the proximal tibia on the medial side, is depicted. On the DECT 3D maps (**B**,**F**) some BME is apparent around the fractures (arrows). On the corresponding DECT coronal 2D images (**C**,**G**), the presence of BME is confirmed (arrows), with a periphery to center gradient. The standard coronal reconstructed CT images (**D**,**H**) confirm the fracture lines (arrows). Note the absence of reactive sclerosis in the left knee (**D**) and the presence of sclerosis in the right one (**H**).

## 10. Inflammatory Disorders

BME represents a common imaging change that is detectable in several inflammatory disorders of the skeletal system, such as autoimmune or septic arthritis. While BME can be used to evaluate the response to therapy in the short term, bone erosions and morphological changes can be used to evaluate the course of disease in the long term (Figures 8 and 9). In cases of advanced inflammatory arthritis, the presence of osteopenia makes it easier for DECT to identify BME, while high resolution CT images can finely identify bone erosions, articular space narrowing and bone remodeling (Figure 8). In painful gout, in addition to identifying tophi, DECT can also assess bone remodeling and detect associated BME [14] (Figure 9). Furthermore, DECT can accurately evaluate BME in patients with sacroiliitis associated with axial spondylarthritis [13].

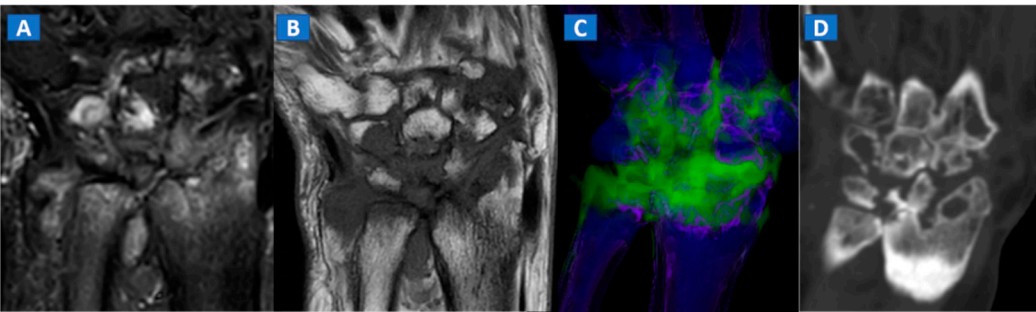

**Figure 8.** A 59-year-old female with advanced rheumatoid arthritis. On the coronal STIR MRI image (**A**), diffuse BME of the wrist and carpal bone is evident as signal hyperintensity. On the corresponding T1-weighted image (**B**), advanced erosive changes are apparent. On the 3D coronal DECT image (**C**), BME is confirmed. The coronal 1 mm standard CT image with soft tissue window (**D**) allows the depiction of erosive and morphological changes of the carpal bones.

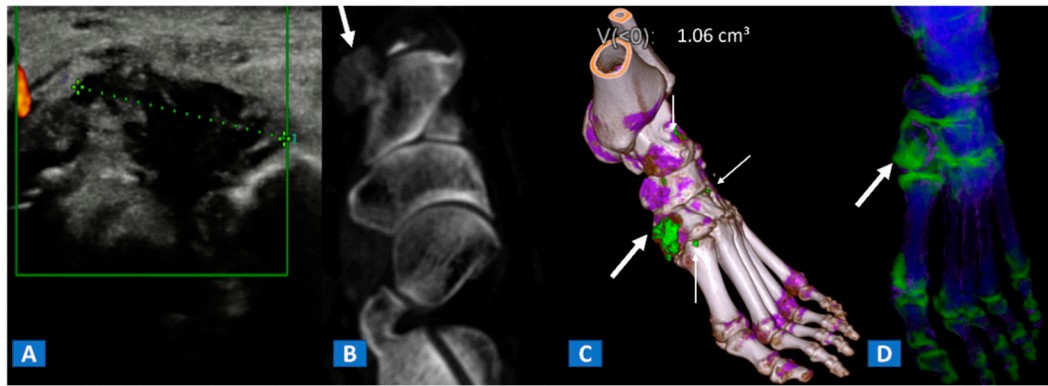

**Figure 9.** A 48-year-old female with painful gout. Upon ultrasound study, a hypoechoic tophus is identifiable (**A**). In the axial CT image with soft-tissue window (**B**), the tophus shows soft-tissue density (arrow). On the 3D DECT image for gout analysis (**C**), the tophus is coded in green (thick arrow). Note the presence of tiny additional tophi (thin arrows). On the DECT 3D map (**D**), BME can be identified on the intermediate cuneiform and on the base of the first metatarsal bone; this may be an additional factor contributing to pain.

## 11. Infectious Diseases

Bone inflammation due to infective diseases involving skeletal system, is often characterized by the presence of BME at imaging. In case of spondylodiscitis, DECT can depict the presence of vertebral bodies and intervertebral disc oedema (Figure 10), which can rule out endplate erosion or vertebral collapse. Furthermore, CT can be used to guide sampling during paraspinal fluid collections. In case of osteomyelitis, DECT could be employed to depict BME, but also to clearly depict erosive changes, lytic lesions, as well as soft tissue involvement.

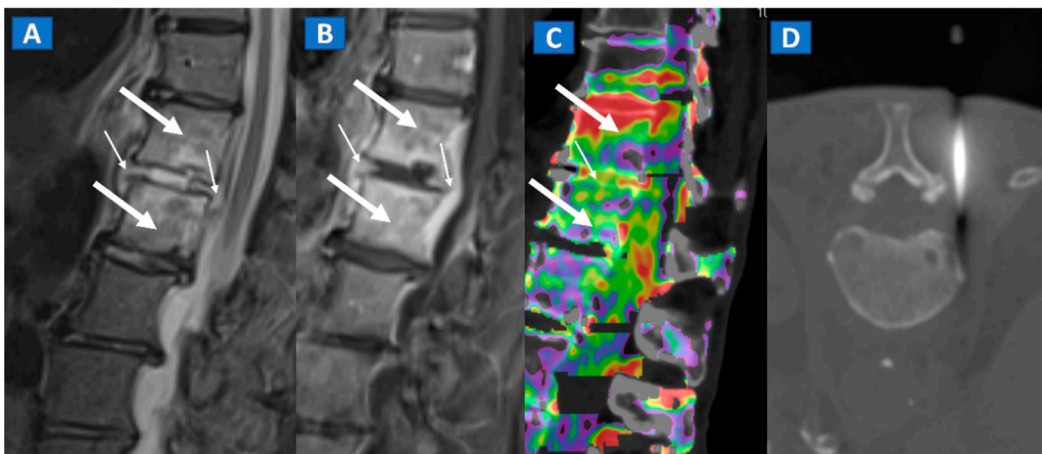

**Figure 10.** A 57-year-old male diagnosed with spondylodiscitis, and presenting with worsening low back pain. On the sagittal STIR MRI image (**A**), BME of the L1 and L2 vertebral bodies is apparent (thick arrows). The intervertebral disc appears hyperintense, with thickening of the anterior and posterior longitudinal ligaments (thin arrows). On the corresponding sagittal T1-weighted image (**B**) after intravenous administration of gadolinium, contrast enhancement is depicted, involving vertebral bodies (thick arrows) and thickened anterior and posterior longitudinal ligaments (thin arrows). On the DECT sagittal 1 mm reconstructed image (**C**), there is BME involving vertebral bodies (thick arrows) and the disc (thin arrow). Under CT guidance (**D**), sampling on para-spinal fluid collection was performed to confirm the diagnosis of spondylodiscitis and to isolate the pathogen involved.

## 12. Myeloma and Bone Metastases

By using VNCa numbers, DECT, as compared to MRI, can reliably depict plasma cell disorder imaging patterns [15]. DECT therefore represents a promising tool for evaluating and staging lymphoproliferative disease. Finally, DECT could help in the diagnosis of CT occult bony lesions and their subsequent biopsies [17] (Figure 11).

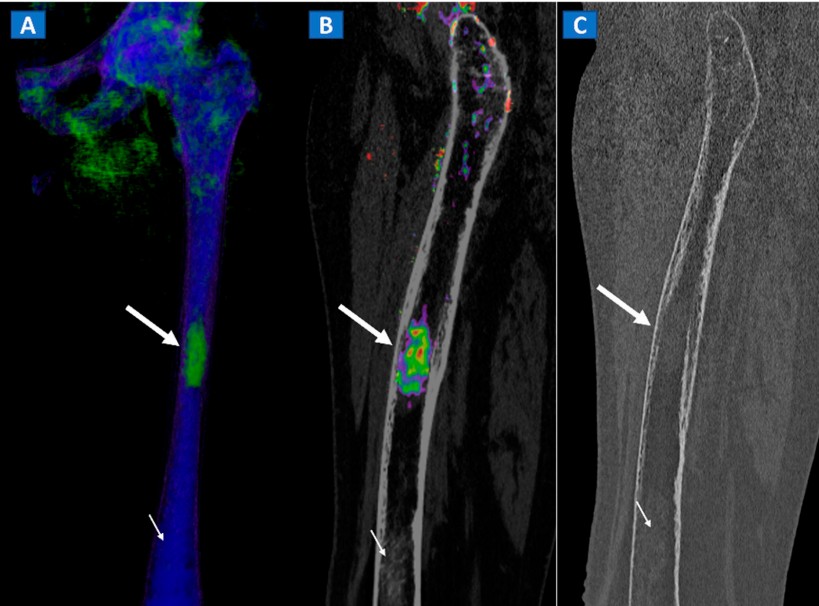

**Figure 11.** A 49-year-old female with single occult femoral metastasis from breast cancer. On the 3D DECT image (**A**) a focal area of edema can be clearly recognized on the left femur (thick arrow). On the para-sagittal 2D image, the lesion (thick arrow) is confirmed on the super-imposed BME map (**B**). The lesion (thick arrow) is not recognized on the para-sagittal 2D bone window image (**C**), showing only a non-specific mildly hyperdense pattern, present also on distal femur (thin arrow), and not associated with BME on the DECT images.

## 13. Conclusions

DECT represents a readily available, artifact-free imaging tool capable of identifying BME and associated bony and soft tissue imaging findings in several non-traumatic settings.

**Author Contributions:** Conceptualization: G.F., G.S. and C.Z.; data curation: G.F. and S.G.; formal analysis: G.F., G.B. and S.M.; investigation: G.F.; methodology: G.F. and S.M.; project administration: G.F.; resources: G.S. and V.I.; supervision: S.G., V.I. and C.Z.; validation: G.F.; visualization: G.F.; writing—original draft: G.F., G.B., S.M., S.G., V.I. and C.Z. All authors have agreed to the published version of the manuscript.

**Funding:** This research received no external funding.

**Institutional Review Board Statement:** All procedures performed in studies involving human participants were in accordance with the ethical standard of the institutional and/or national research committee and with the 1964 Helsinki declaration and its later amendments or comparable ethical standard.

**Informed Consent Statement:** Informed consent was waived for this pictorial study.

**Data Availability Statement:** Images and cases were acquired by the authors at IRCCS Saco Cuore Don Calabria Hospital.

**Acknowledgments:** The manuscript was edited by Elinor Julie Rae Anderson.

**Conflicts of Interest:** The authors declare that they have no conflicts of interest.

## Abbreviations

| | |
|---|---|
| DECT | Dual-energy computed tomography |
| BME | bone marrow oedema |
| MRI | magnetic resonance imaging |
| OCL | osteochondral lesions |
| AVN | avascular necrosis |
| VNCa | virtual non-calcium |
| FOV | field of view |
| ROI | region of interest |

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
