# Peer review of "Identification of Non-Traumatic Bone Marrow Oedema: The Pearls and Pitfalls of Dual-Energy CT (DECT)"

_tomography, doi:10.3390/tomography7030034_

Round 1

Reviewer 1 Report

DECT is an extensively published topic. The article does not provide additional information. Still, I can see some novelty in VNca algorithms. (Dual-Energy CT in Musculoskeletal Imaging: What Is the Role Beyond Gout? Read More: https://www.ajronline.org/doi/full/10.2214/AJR.19.21095)

MRI is still preferred method to DECT while detecting spondylodiskitis and disc herniations. Detecting disc herniation on soft tissue windows is not specific to DECT as conventional CT soft tissue window can also produce similar images. 

There some merits in using DECT where MRI is not possible such as osteoporotic vertebral fractures and pelvic insufficiency fractures.

This manuscript would be more interesting if presented as a pictorial review in challenging cases and showing DECT as a problem-solver in specific situations rather than suggesting its generic utility in BME diagnosis.

It would be of interest if authors can include use of DECT in diagnosis of CT occult bony lesions and their subsequent biopsies.

Reviewer 2 Report

TITLE

Identification of non-traumatic bone marrow oedema: The pearls and pitfalls of dual-energy CT (DECT)

This is an interesting pictorial manuscript focusing on the role of DECT in diagnosing non-traumatic bone marrow edema. Some diagnostic applications were recently described in the literature; however, this is the first paper including multiple applications of DECT in clinical practice. This paper could represent a starting point for future studies that may confirm the role of DECT in non-traumatic MSK imaging.

ABSTRACT, KEY WORDS AND ABBREVIATIONS

Fine.

INTRODUCTION

The background is briefly described. However, some additional details regarding the evaluation of non-traumatic pain/bone marrow edema with DECT could be included here. For example, regarding the identification of BME in sacroiliitis and inflammatory spine disease. Also some references regarding the identification of disk hernia should be included here.

IMAGE PROTOCOL AND INTERPRETATION

Fine.

VERTEBRAL COMPRESSION FRACTURES

The topic is briefly described. Some details regarding recently published papers could help the readers to focus on the problem. Also, some additional references should be added.

NON-TRAUMATIC OSTEOCHONDRAL LESIONS

Fine.

AVASCULAR NECROSIS

Some details regarding the staging of disease should be included.

TRANSIENT BONE MARROW OEDEMA SYNDROME

This paragraph should include some details regarding the differential diagnosis.

STRESS OR INSUFFICIENCY FRACTURES

Fine.

INFLAMMATORY DISORDERS and MYELOMA AND BONE METASTASES

I would give some additional details and citations regarding these topics.

INFECTIOUS DISEASES

Fine

References

Add additional recent citations regarding the aforementioned topics.

Figures

Interesting explicative cases with nice DECT-MRI correlation.

Fig 2 is not labelled.

Author Response

Dear Editor and dear Reviewers

This is a point to point rebuttal letter regarding the review of the manuscript entitled

Identification of Non-Traumatic Bone Marrow Oedema: The Pearls and Pitfalls of Dual-Energy CT (DECT)

Reviewer 1

DECT is an extensively published topic. The article does not provide additional information. Still, I can see some novelty in VNca algorithms. (Dual-Energy CT in Musculoskeletal Imaging: What Is the Role Beyond Gout? Read More: https://www.ajronline.org/doi/full/10.2214/AJR.19.21095)

Authors Thank you for your comment. Only some topics were described in the above mentioned paper. Still, we agree with you that our paper may bring some novelty.

We will include the suggested citation in the text and in the reference list

Rajiah P, Sundaram M, Subhas N. Dual-Energy CT in Musculoskeletal Imaging: What Is the Role Beyond Gout? AJR Am J Roentgenol. 2019 Sep;213(3):493-505. doi: 10.2214/AJR.19.21095. Epub 2019 Apr 30. PMID: 31039024.

Reviewer 1

MRI is still preferred method to DECT while detecting spondylodiskitis and disc herniations. Detecting disc herniation on soft tissue windows is not specific to DECT as conventional CT soft tissue window can also produce similar images. 

Authors Thank you for your comment. We agree that MRI is still preferred method for diagnosing disk herniation. However, we believe that DECT may bring some additional contrast with respect to conventional CT, with pontetial benefit in difficult cases.

Reviewer 1

There some merits in using DECT where MRI is not possible such as osteoporotic vertebral fractures and pelvic insufficiency fractures.

This manuscript would be more interesting if presented as a pictorial review in challenging cases and showing DECT as a problem-solver in specific situations rather than suggesting its generic utility in BME diagnosis.

Authors Thank you for your comments. The paper is indeed presented as a pictorial review showing interesting and explicative cases. For this purpose, we presented several cases in which the detection of BME by DECT could represent the key for diagnosis. A case of an occult single bone metastatis detected only by DECT was included, as suggested.

Reviewer 1

It would be of interest if authors can include use of DECT in diagnosis of CT occult bony lesions and their subsequent biopsies.

Authors Thank you for your comment. We will include and additional case of occult bony lesion; the last paragraph will be changed accordingly

Figure 11. A 49-year-old with single occult femoral metastasis from breast cancer. On the 3D DECT image (A) a focal area of edema can be clearly recognized on the left femur (thick arrow). On the para-sagittal 2D image, the lesion (thick arrow) is confirmed on the super-imposed BME map (B). The lesion (thick arrow) is not recognized on the para-sagittal 2D bone window image, showing only a non-specific mildly hyperdense pattern, present also on distal femur (thin arrow), and not associated with BME on the DECT images.

Reviewer 2

Identification of non-traumatic bone marrow oedema: The pearls and pitfalls of dual-energy CT (DECT)

This is an interesting pictorial manuscript focusing on the role of DECT in diagnosing non-traumatic bone marrow edema. Some diagnostic applications were recently described in the literature; however, this is the first paper including multiple applications of DECT in clinical practice. This paper could represent a starting point for future studies that may confirm the role of DECT in non-traumatic MSK imaging.

Authors Thank you for your comment.

ABSTRACT, KEY WORDS AND ABBREVIATIONS

Fine.

Reviewer 2

INTRODUCTION

The background is briefly described. However, some additional details regarding the evaluation of non-traumatic pain/bone marrow edema with DECT could be included here. For example, regarding the identification of BME in sacroiliitis and inflammatory spine disease. Also some references regarding the identification of disk hernia should be included here.

Authors Thank you for your comment.

The background regarding inflammation and DECT was changed according to the R2 suggestions, as follows:

MRI is the most frequently used imaging tool in the diagnosis of BME [1]. However, because of contraindications and increased costs, it is not always suitable or available. DECT can suppress normal bone by using a virtual non-calcium (VNCa) algorithm, thus allowing the identification of BME in traumatic and non-traumatic settings [6–17]. Specifically, DECT can be used to diagnose such non-traumatic pathological conditions as gout. In particular, DECT is very useful in atypical presentations, which pose a diagnostic challenge and distinguish it from other arthropathies and masses, such as septic arthritis, rheumatoid arthritis, osteoarthritis, pseudogout, or tumor, especially in sites that are not directly accessible for fluid aspiration [17].

Reviewer 2

IMAGE PROTOCOL AND INTERPRETATION

Fine.

Reviewer 2

VERTEBRAL COMPRESSION FRACTURES

The topic is briefly described. Some details regarding recently published papers could help the readers to focus on the problem. Also, some additional references should be added.

Authors Thank you for your comment.

The paragraph regarding vertebral fractures was changed including a new citation, as follows:

DECT can distinguish fresh from old vertebral compression fractures by demonstrating the presence of BME [11]. A recent metanalysis evaluating 13 studies using MRI as reference for diagnosis, including 515 patients and 926 acute fractures, DECT overall sensitivity was 86.2% with a specificity of 91.2% and accuracy of 89.3% [18]. In addition, DECT can accurately determine the position of hardware inserted during spinal surgery, by reducing metallic artifacts. The high intrinsic contrast of DECT images also assists in determining a differential diagnosis, such as in cases of disc herniation (figure 2).

NON-TRAUMATIC OSTEOCHONDRAL LESIONS

Fine.

AVASCULAR NECROSIS

Some details regarding the staging of disease should be included.

Authors: as suggested, the following sentence was added.

 Thanks to high resolution images, DECT could be used for staging AVN, depicting for example cystic and sclerotic radiographic changes in early phases, or BME around sub-chondral collapsed areas.

TRANSIENT BONE MARROW OEDEMA SYNDROME

This paragraph should include some details regarding the differential diagnosis.

Authors: as suggested, the following sentence was added.

DECT is indeed capable to diagnose stress fractures, early AVN stages, inflammatory arthropathies.

STRESS OR INSUFFICIENCY FRACTURES

Fine.

INFLAMMATORY DISORDERS and MYELOMA AND BONE METASTASES

I would give some additional details and citations regarding these topics.

Authors: as suggested, the following paragraph was changed. In particular, a sentence and a dedicated figure was included, as suggested by R1 and R2.

By using VNCa numbers, DECT, as compared to MRI, can reliably depict plasma cell disorder imaging patterns [15]. DECT therefore represents a promising tool for evaluating and staging lymphoproliferative disease. Finally, DECT could help in the diagnosis of CT occult bony lesions and their subsequent biopsies [17] (fig 11).

INFECTIOUS DISEASES

Fine

References

Add additional recent citations regarding the aforementioned topics.

Authors: as suggested, additional recent citations were included in the reference list

Figures

Interesting explicative cases with nice DECT-MRI correlation.

Fig 2 is not labelled.

Authors: as suggested, a labelled figure 2 was included.

Round 2

Reviewer 1 Report

I would like to thank authors extensively edit the manuscript and putting efforts to make more interesting